# VIRify: An integrated detection, annotation and taxonomic classification pipeline using virus-specific protein profile hidden Markov models

**Guillermo Rangel-Pineros**[1,2,3]*, **Alexandre Almeida**[3,4,5], **Martin Beracochea**[3], **Ekaterina Sakharova**[3], **Manja Marz**[6,7], **Alejandro Reyes Muñoz**[2], **Martin Hölzer**[3,7,8], **Robert D. Finn**[3]*

**1** The Globe Institute, Faculty of Health and Medical Sciences, University of Copenhagen, Copenhagen, Denmark, **2** Max Planck Tandem Group in Computational Biology, Department of Biological Sciences, Universidad de los Andes, Bogota, Colombia, **3** European Molecular Biology Laboratory, European Bioinformatics Institute (EMBL-EBI), Wellcome Genome Campus, Hinxton, United Kingdom, **4** Wellcome Sanger Institute, Wellcome Genome Campus, Hinxton, United Kingdom, **5** Department of Veterinary Medicine, University of Cambridge, Cambridge, United Kingdom, **6** RNA Bioinformatics, Friedrich Schiller University, Jena, Germany, **7** European Virus Bioinformatics Center, Friedrich Schiller University, Jena, Germany, **8** Methodology and Research Infrastructure, Genome Competence Center (MF1), Robert Koch Institute, Berlin, Germany

* guillermo.pineros@sund.ku.dk (GRP); rdf@ebi.ac.uk (RDF)

**Data Availability Statement:** The metagenomic assemblies generated for the selected mock communities and the set of informative ViPhOGs are available on OSF at https://doi.org/10.17605/

## Abstract

The study of viral communities has revealed the enormous diversity and impact these biological entities have on various ecosystems. These observations have sparked widespread interest in developing computational strategies that support the comprehensive characterisation of viral communities based on sequencing data. Here we introduce VIRify, a new computational pipeline designed to provide a user-friendly and accurate functional and taxonomic characterisation of viral communities. VIRify identifies viral contigs and prophages from metagenomic assemblies and annotates them using a collection of viral profile hidden Markov models (HMMs). These include our manually-curated profile HMMs, which serve as specific taxonomic markers for a wide range of prokaryotic and eukaryotic viral taxa and are thus used to reliably classify viral contigs. We tested VIRify on assemblies from two microbial mock communities, a large metagenomics study, and a collection of publicly available viral genomic sequences from the human gut. The results showed that VIRify could identify sequences from both prokaryotic and eukaryotic viruses, and provided taxonomic classifications from the genus to the family rank with an average accuracy of 86.6%. In addition, VIRify allowed the detection and taxonomic classification of a range of prokaryotic and eukaryotic viruses present in 243 marine metagenomic assemblies. Finally, the use of VIRify led to a large expansion in the number of taxonomically classified human gut viral sequences and the improvement of outdated and shallow taxonomic classifications. Overall, we demonstrate that VIRify is a novel and powerful resource that offers an enhanced capability to detect a broad range of viral contigs and taxonomically classify them.

OSF.IO/FBRXY. All the scripts used for the manual curation of the ViPhOGs and the Nextflow implementation of VIRify are available at https://github.com/EBI-Metagenomics/emg-viral-pipeline.

**Funding:** This work was supported by the Deutsche Forschungsgemeinschaft (CRC 1076 AquaDiva to MH), and the Biotechnology and Biological Sciences Research Council (BB/P027849/1 CABANA project for capacity building for bioinformatics in Latin America to GRP). The funders had no role in study design, data collection and analysis, decision to publish, or preparation of the manuscript.

**Competing interests:** The authors have declared that no competing interests exist.

## Author summary

Viruses are the most abundant biological entities on our planet. Some are relevant pathogens for public health or agriculture. Still, many also play ecological roles that are critical for maintaining ecosystems. Most viruses are yet to be cultured, so their identification and characterisation depend solely on the analysis of DNA or RNA obtained from the environment. Unlike cellular organisms, viruses also lack a universal genetic marker that allows taxonomic profiling of an environmental viral community. We have manually curated a set of specific viral protein models that serve as taxonomic markers for a comprehensive range of viral taxa. Using these protein models, we developed VIRify, a computational pipeline for the detection, annotation, and taxonomic classification of viral sequences obtained from environmental DNA or RNA. Our new pipeline was efficient in detecting and classifying sequences of viruses targeting bacteria or eukaryotic organisms in mock microbial communities, samples from the world's oceans, and a previously assembled collection of human gut viruses. VIRify is user-friendly, requires minimal interaction with the command line, and was developed with portability in mind. VIRify can enhance the exploration of viral diversity in nature and support the detection of pathogenic viruses with pandemic potential.

## Introduction

Viruses are the most abundant biological entities inhabiting our planet, with an estimated $10^{30}$ virus-like particles (VLP) present in the world's oceans [1]. Even though viruses are often acknowledged as disease-causing agents in animals and plants, many studies to date have demonstrated the pivotal role that viruses play in shaping microbial populations, particularly in the case of phages and their bacterial hosts [2–5]. The application of metagenomics-based approaches in these studies facilitated the culture-independent exploration of microbial communities, resulting in the expansion of the span of viral-host interactions that could be characterised. This approach has revealed a vast extent of uncharacterised viral genetic diversity known as "viral dark matter" [6–11] and led to the discovery of numerous uncultivated virus genomes (UViGs) [12–15]. Studies, especially in oceanic environments, have demonstrated the importance of viruses in biogeochemical cycling and metabolic modulation of ocean microbes [12,16,17].

A review paper of the TARA Oceans project highlighted the urgent need for automated approaches to systematically organise the enormous virus sequencing data beyond the species level [18]. Several computational tools and resources that offer a solution to this challenge have been developed and made publicly available in the last two decades [19–21]. One of the most prominent and currently available tools is VConTACT2, which employs data on proteins shared between phage genomes to build a similarity network that is further analysed to identify clusters of evolutionarily related phages [22]. An alternative approach to the taxonomic classification of viruses involves protein profile HMMs representing clusters of homologous proteins or protein domains, which in turn serve as markers for different viral taxa, a strategy used for example by ClassiPhage [23]. Phage classification with ClassiPhage demonstrated agreement with the reference taxonomy from the International Committee on Taxonomy of Viruses (ICTV), although the generation and testing of the taxon-specific profile HMMs was solely focused on phages targeting members of the bacterial family *Vibrionaceae* [23]. More recently, a set of 31,150 profile HMMs representing proteins predicted in viral genomes from

NCBI were used as the basis for a set of Random Forest classifiers aimed at assigning viral taxonomy at the order, family, and genus ranks [24]. These classifiers demonstrated very high classification accuracy ($\geq$ 89%) at all the aforementioned ranks, suggesting that these profile HMMs (or ViPhOGs as they were named) could be the foundation for a new method that taxonomically classifies viral genomic sequences present in metagenomic datasets.

Existing pipelines such as VirMiner, iVirus and KBase enable taxonomic and functional characterisation of viral contigs in metagenomic datasets [25–27]. More recently, Hecatomb and MetaPhage were released to provide the scientific community with scalable and portable resources for the taxonomic profiling of viral communities, using metagenomic reads as input [28,29]. Here we present VIRify, a new pipeline that predicts virus-like sequences using canonical viral signals and annotates contigs using a comprehensive collection of profile HMMs, including our set of manually curated ViPhOGs. VIRify reduces false-positive predictions and prioritises portability via its implementation as a containerised Nextflow workflow [30], which can be directly integrated into already existing platforms such as MGnify [31]. The VIRify pipeline is freely available at https://github.com/EBI-Metagenomics/emg-viral-pipeline. We showcase the application of VIRify based on two viral mock communities mostly comprised of phages [32] but also eukaryotic viruses [33], as well as 243 metagenome assemblies of the TARA Oceans project [12], and a previously generated collection of 57,866 viral genomic clusters (VCs) from globally-distributed human gut metagenomic datasets [34].

## Methods

### Manual curation of virus-specific protein HMMs and definition of model-specific bit score thresholds

We manually curated a set of taxon-specific profile HMMs by searching the original ViPhOG database [24] against all entries in UniProtKB (February 2019 version). This was achieved using hmmsearch (v3.2.1) and setting a per-sequence reporting e-value threshold of $1.0 \times 10^{-3}$. The resulting output was analysed using in-house python scripts (https://github.com/EBI-Metagenomics/emg-viral-pipeline/tree/master/bin/models_vs_uniprot_check) to identify which ViPhOGs could be used as taxon-specific markers, hereafter referred to as informative ViPhOGs. These were identified after applying the following steps to the hmmsearch output of each ViPhOG:

1. Reported taxonomy identifiers (taxid) were recorded, along with the highest bit score obtained for each one of them and their associated taxa at the genus, subfamily, family and order ranks.

2. For each one of the target taxa recorded in the previous step, the corresponding highest bit scores were sorted and a bit score range was defined using the lowest and highest of these values.

3. Starting with the target taxa at the genus rank, it was determined whether the query ViPhOG was associated with a single taxon or the bit score range for the best taxon did not overlap with the range obtained for the remaining taxa. If either of these conditions was fulfilled, then the query ViPhOG was selected as informative at the genus rank. Otherwise, the procedure was repeated at the subfamily, family and order ranks, until the query ViPhOG could be selected as informative for any of them. If the query ViPhOG was not assigned to any taxon at the searched ranks, it was considered non-informative.

To leverage the data obtained from the analysis described above and set inclusion bit score thresholds suitable for each model, we defined two bit score thresholds (S1 and S2) for each

informative ViPhOG. S1 was defined as the minimum bit score associated with the ViPhOG's target taxon, whereas S2 was set as the maximum bit score obtained for other viral taxa identified through the previously described procedure. For the cases in which no further taxa were reported at the same rank as the target taxon, only S1 was set for the corresponding informative ViPhOG. Next, we define GA (gathering), TC (trusted cutoff), and NC (noise cutoff) parameters of each informative ViPhOG model. GA thresholds define reliable, curated thresholds for family memberships, NC thresholds describe the highest-scoring known false positive, and TC thresholds refer to the score of the lowest-scoring known true positive that is above all known false positives, which reflects a similar strategy to Pfam [35]. In addition, each parameter (GA, TC, NC) defines two thresholds for reporting and inclusion scores: the per-sequence threshold ($GA_{seq}$, $TC_{seq}$, $NC_{seq}$) and the per-domain threshold ($GA_{dom}$, $TC_{dom}$, $NC_{dom}$). We used our previously defined bit score thresholds S1 and S2 to set these six parameters specifically for each informative ViPhOG model as follows:

$$GA_{seq}, TC_{seq} = S1$$

$$GA_{dom}, TC_{dom} = S1 - 3 \ if \ (S1 - 3) > S2, else \ S1$$

$$NC_{seq}, NC_{dom} = S2$$

S1 was added as TC and GA per-sequence threshold, and trimmed by three bit to apply a per-domain threshold. Trimming was only applied if S1 did not drop below S2. The S2 bit score was unmodified and used as both per-sequence and per-domain NC. For the cases in which no S2 bit score was recorded, the noise cutoff was omitted and the sequence-specific S1 value was trimmed by three bits to allow less restrictive predictions. Finally, the values set for the GA, TC, and NC parameters were added to the header section of each informative ViPhOG's HMM file.

## Coverage of current viral taxonomy and calculation of taxon-specific ViPhOG-to-CDS ratios

The set of informative ViPhOGs was screened to keep the ones that were associated with valid taxa in the latest release of ICTV's viral taxonomy MSL#38. These were subsequently employed to determine the extent to which the lineages comprising the current viral taxonomy are covered by our set of informative ViPhOGs. A cladogram that included all viral genera and corresponding ancestral taxa in NCBI's Taxonomy database from January 2023 was built using the ete3 python package [36,37]. The number of informative ViPhOGs identified for each viral genus was mapped on the cladogram using the interactive Tree of Life (iTOL) resource and the associated annotation file for plotting numeric data as bars [38]. In addition, taxa in the subfamily, family and order ranks for which at least one informative ViPhOG had been identified were highlighted using iTOL's annotation file for labelling nodes with symbols (**S1 Fig**).

Viral taxa linked to the informative ViPhOGs were categorized as prokaryotic or eukaryotic based on their target host. Currently known viral-host relationships were retrieved from Virus-Host DB (release 214 from November 2022), which contained 12,032 viral entries targeting eukaryotes and 5,491 viral entries targeting either bacteria or archaea [39].

For each taxon associated with any of the informative ViPhOGs or that is part of the viral lineages covered by them, we determined the ratio between the number of associated informative ViPhOGs and the average number of CDS. To calculate this ratio, the number of informative ViPhOGs for taxa in the subfamily, family and order ranks included the models linked to the corresponding descendant taxa, in addition to those directly linked to them. To determine

the average number of CDS per taxon, we identified all viral assemblies from NCBI's Assembly database that had been deposited until November 2022 and whose CDS had been annotated. We retrieved the corresponding assembly report files and used them to extract the taxids and number of CDS per assembly. The taxids were employed to retrieve the taxa at the genus, sub-family, family and order ranks included in the lineage of each assembly, using the ete3 python package. For each one of these taxa, the average number of CDS and corresponding standard deviation were calculated based on the assemblies belonging to each one of them. Finally, we calculated the ratio between the number of informative ViPhOGs associated with each taxa and the corresponding average number of CDS, which was ultimately defined as taxon-specific ratio (TSR). The value of this parameter was set to 1 for the cases in which the number of informative ViPhOGs was higher than the average number of CDS.

## Preparation of mock metagenome assemblies for benchmarking

Paired-end reads of six virus-enriched samples were downloaded from ENA (study: PRJEB6941; runs: ERR575691, ERR575692, ERR576942, ERR576943, ERR576944, ERR576945) and concatenated to make a single input file [32]. Before assembly, host contamination was removed by k-mer-based decontamination against the mouse genome (Ensembl GRCm38, primary assembly) with BBDuK v38.79 from the BBTools suite (https://sourceforge.net/projects/bbmap/) using a k-mer size of 27. To include a metagenome assembly that also comprised eukaryotic viruses, we further obtained paired-end reads of seven virus-enriched samples (study: PRJNA319556; runs: SRR3458563-3458569) [33].

For the combined data sets within both studies, two individual metagenome assemblies were performed by first cleaning the reads with fastp v0.20.0 [40] and passing them to metaS-PAdes v3.14 [41] using default parameters. The '—only-assembler' parameter was used to generate the assembly of the combined Neto read sets due to memory constraints. We used QUAST v5.0.2 [42] to assess basic quality metrics of the assemblies. The two assemblies (here-after named Kleiner and Neto assemblies) can be downloaded from the Open Science Framework (https://doi.org/10.17605/OSF.IO/FBRXY).

## Selection and comparison of virus prediction tools

We used the multi-tool workflow "What the Phage" (WtP) [43] for comparing different virus prediction tools in an attempt to identify the best tool combination for a comprehensive initial virus prediction of our pipeline. We run WtP release v0.9.0, including VirSorter (with and without virome option), VirFinder (default and VF.modEPV_k8.rda models), PPR-Meta, DeepVirFinder [44], MARVEL [45], metaPhinder [46], VIBRANT [47], VIRNET [48], Phi-garo [49] and sourmash [50] on the Kleiner and Neto assemblies. The analysis was performed on contigs that were at least 1.5 kb long to filter out shorter contigs with potentially few ORFs, which are less likely to be correctly classified by our ViPhOG-based taxonomic annotation pipeline. Previous studies have revealed that viral genomes shorter than 1.5 kb tend to code for no more than 5 proteins [24]. Based on a previous study in which global oceanic viral populations were surveyed, we manually updated the parameter configuration file of WtP to include all VirSorter predictions from categories 1–5 and to filter VirFinder results by p-values < 0.05 and scores ≥ 0.7 [12].

To assess the performance of each viral prediction tool and the tool combination implemented in VIRify, we identified the viral contigs in the Kleiner and Neto assemblies by aligning them to the genomes of the viruses that comprised the corresponding mock communities. For both mock communities, the set of viral genomes used as targets for the alignments included all the viruses listed in the corresponding studies, and the putative prophages

predicted in the genomes of the bacteria that were either part of the mock communities or used for propagating the mock community phages. Thus, prophage prediction was conducted for the following bacterial genome sequences: *Pseudomonas savastanoi* pv. *phaseolicola* strain HB10Y (GCF_001294035), *Salmonella enterica* subsp. *enterica* serovar Typhimurium str. LT2 (NC_003197), *Listeria monocytogenes* EGD-e (NC_003210), *Bacteroides thetaiotaomicron* VPI-5482 (NC_004663), *Enterococcus faecalis* V583 (NC_004668), *Escherichia coli* B strain C3029 (NZ_CP014269), *Lactobacillus acidophilus* strain ATCC 4356 (GCF_000786395), *Bifidobacterium animalis* subsp. *lactis* ATCC 27674 (GCF_001263985), *B. thetaiotaomicron* strain ATCC 29741 (GCF_004349615) and *E. coli* DSM 30083 (NZ_CP033092). Prophages were predicted from the listed bacterial genomes using VirSorter [51], selecting the Viromes database and activating the virome decontamination mode, and PHASTER [52].

VirSorter predictions in categories 4 and 5, and PHASTER predictions classified as Intact or Questionable were clustered using cd-hit-est with a sequence identity cut-off of 0.9 and default settings [53]. The set of putative prophages selected for each mock community included all VirSorter category 4 predictions, PHASTER Intact prophages, and regions that were reported both as VirSorter category 5 predictions and PHASTER Questionable prophages. Contigs from each mock community assembly were aligned to the corresponding set of viral genomes using nucmer v3.23 with default settings [54], and contigs were identified as viral when at least 70% of their length was covered by an alignment with at least 90% sequence identity. Following the identification of viral contigs within the Kleiner and Neto assemblies, the performance of the viral predictions tools tested using WtP and the combination selected for VIRify was measured via the calculation of the F1-score (**S2 Fig**).

The output from WtP was visualised in UpSet plots and presence/absence maps that indicated whether an input contig was predicted as viral or not by each of the assessed tools (**S3 Fig**). MARVEL and VIRNET predictions were not included for the Neto assembly because both tools failed to process this assembly for unidentified technical reasons.

## VIRify taxonomic annotation pipeline description

**Identification of putative viral sequences.**   Based on the results obtained with WtP and the procedure previously employed in a global oceanic survey of viral populations, it was determined that VIRify's detection of putative viral contigs would be carried out with VirFinder, VirSorter and PPR-Meta, while putative prophage detection would be conducted with VirSorter only (**Fig 1**) [12,51,55,56]. Viral prediction with VirSorter is carried out using parameters '—db 2' and '—virome' when the processed metagenomic assemblies are generated from virome datasets (as was the case for VIRify's benchmarking with the Kleiner and Neto assemblies), otherwise the latter parameter is not set. Putative prophages are retrieved from VirSorter predictions in prophage categories 4 and 5 as defined by the tool. Detection of putative viral contigs with VirFinder is conducted using the classifiers in the model file VF.modEPV_k8.rda, which was trained using sequences from prokaryotic and eukaryotic viruses (available at https://github.com/jessieren/VirFinder). Predicted viral contigs are classified into two different categories within the VIRify pipeline as follows: contigs reported by VirSorter in categories 1 and 2 are placed in the high-confidence category, whilst the low-confidence category includes contigs that satisfy any of the following conditions:

- Reported by VirFinder with $p < 0.05$ and $score \geq 0.9$.

- Reported by VirFinder with $p < 0.05$ and $score \geq 0.7$, and also reported by VirSorter in category 3.

- Reported by VirFinder with $p < 0.05$ and $score \geq 0.7$, and also reported as phage by PPR-Meta.

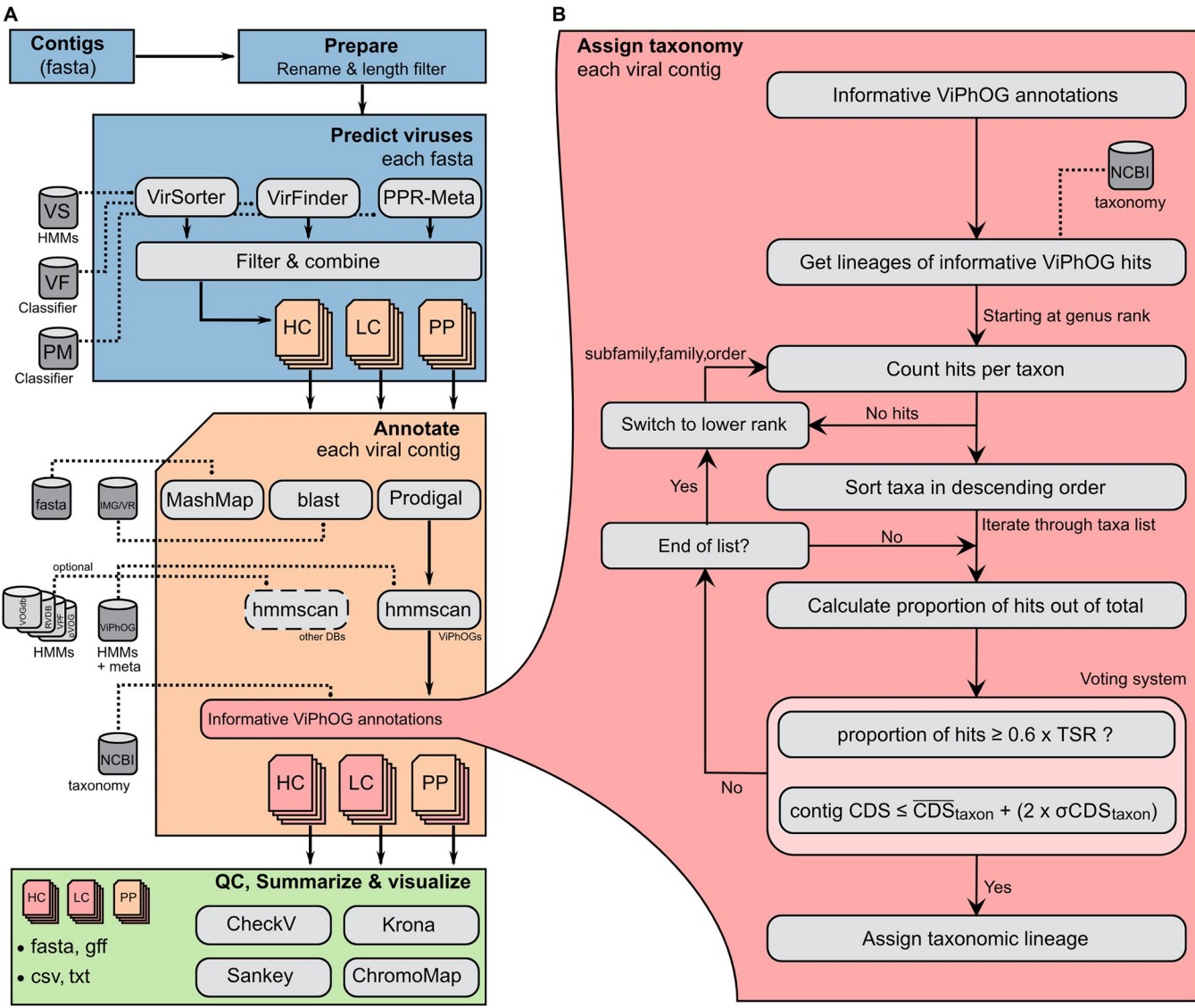

**Fig 1. Overview of the VIRify pipeline. (A)** Starting from a set of contigs (fasta file) the pipeline preprocesses the input sequences (ID renaming, length filtering) and predicts contigs from a putative viral origin that are split into high confidence (HC), low confidence (LC) and putative prophage (PP) sets. Each selected contig is then annotated and assigned to a taxonomy, if possible. All results (annotated viral contigs) are quality-controlled with CheckV, and finally summarised and visualised. **(B)** The assigned taxonomy is based on the informative ViPhOG hits per contig and performed on genus, family, subfamily and finally order rank. We consider high-confidence and low-confidence ViPhOG hits and discard non-informative models where no clear taxonomy signal could be assigned. TSR—taxon-specific ratio; $\overline{CDS}_{taxon}$—taxon average CDS; $\sigma CDS_{taxon}$—taxon CDS standard deviation.

Finally, CheckV [57] is employed to assess the quality and completeness of the categorised viral genome contigs.

**Annotation of putative viral sequences with informative ViPhOGs.** Protein-coding sequences within putative viral contigs and prophages are detected with Prodigal v2.6.3 [58], using the standard bacterial/archaeal translation table (-g 11) and the metagenomic prediction mode (-p meta) (Fig 1). A recent study in which the performance of a range of ORF-prediction tools was assessed for the viral genomes available in NCBI's RefSeq reported that Prodigal was the most accurate tool for DNA viruses, which account for ~97% of the viral genomic

sequences currently available in public databases [59,60]. Amino acid sequences derived from the predicted coding sequences are scanned with the complete ViPhOG database using hmmscan v3.2.1 [61] and the model-specific bit score thresholds defined as previously described, to supersede any thresholding based on statistical significance alone. To apply the defined model-specific thresholds, we set the—cut_ga option during execution of the hmmscan command. However, to account for the set of ViPhOGs for which no bit score thresholds were defined (and therefore set to 0), an additional post-processing filter is applied to the full-sequence e-values reported in the hmmscan output ($E{\leq}1.0{\times}10^{-3}$).

**Taxonomic assignment of putative viral sequences.**   In-house python scripts (see https://github.com/EBI-Metagenomics/emg-viral-pipeline) are employed for parsing the hmmscan results and to provide a taxonomic assignment for all putative viral contigs and prophages, based on the reported hits against the ViPhOG database. Starting at the genus rank, the algorithm checks all taxa associated with the reported informative ViPhOGs, sorts them in descending order based on the proportion of corresponding hits, and iterates through them until it identifies a taxon that fulfils the conditions set in our voting system. The first of these conditions is that the proportion of hits for a candidate taxon must be equal to or higher than 0.6 x TSR. The default value of 0.6 sets a maximum threshold for the number of hits required to call a taxon, but the TSR leads to a reduction in the threshold value that is consistent with the density of informative ViPhOGs linked to each taxon. The second condition checks whether the number of CDS in the contig is not significantly larger than the average number of CDS calculated for the candidate taxon (i.e. $CDS_{contig} \leq \bar{CDS}_{taxon} + 2*\sigma CDS_{taxon}$). This condition was set to reduce the number of wrong taxonomic assignments, as it prevents the classification of contigs into taxa where the average number of CDS is significantly lower than the contig CDS count. If either of the conditions is not satisfied, the algorithm continues the iteration over the remaining taxa until it finds one that satisfies them. If the iteration finishes without success, the algorithm shifts to the next taxonomic rank and repeats the described procedure. The process is repeated at the subfamily, family, and order ranks until a taxon that satisfies the voting system's conditions is found, in which case the algorithm reports the corresponding viral lineage (Fig 1).

**Visualization of assigned viral taxonomies and ViPhOG hits.**   Putative viral contigs with and without an assigned taxonomy are visualized using interactive Krona [62] and Sankey plots inspired by the Pavian package [63]. ORFs identified with Prodigal and their corresponding informative ViPhOG hits are visualized for each contig using the ChromoMap package v0.2 [64]. The package does not visualize exact start and stop positions for each ORF but instead relies on a more general grid view, thus visualizing the general coverage of annotated ORFs for each contig.

## Analysis of 243 TARA Oceans assemblies with VIRify

We obtained 243 ocean microbiome assemblies from ENA (https://www.ebi.ac.uk/ena/browser/view/PRJEB22493), generated as part of the TARA Oceans microbiome study [12]. The FASTA identifiers include the size fraction of the corresponding sample; thus samples enriched for viruses are labeled with the suffix _0.1–0.22. All assemblies were filtered to retain contigs that were at least 5 kb long and these were screened for viruses using VIRify and setting the—virome option to activate VirSorter's virome decontamination mode.

## Taxonomic classification of 57,866 VCs from the Gut Phage Database (GPD)

The GPD is a collection of phage genomic sequences that includes prophages from cultured gut bacteria and putative phage sequences retrieved from assemblies of globally-distributed

human gut metagenomes [34]. A representative sequence for each of the 57,866 VCs in the GPD was selected by choosing the corresponding entry with the highest CheckV quality and completion values. The complete set of representative sequences was analysed with VIRify, using the "—onlyannotate" option to skip the viral prediction step and apply the taxonomic classification pipeline to all the input sequences. The results obtained for all the classified representative sequences were collated and summarised in a Sankey plot.

### Selection and comparison of virus-specific protein profile HMM databases

The taxonomic assignment of VIRify relies purely on our own protein profile HMM database, their taxonomy, and model-specific bit score cutoffs. In an effort to be comprehensive, our pipeline also provides the option of additional annotations by incorporating 25,399 models from the VOGdb (http://vogdb.org/), 25,281 viral protein family (VPF) models from IMG/VR [60], 9,911 models from RVDB v17.0 [65,66], and 9,518 models from pVOG [67]. VIRify can use all databases to individually annotate proteins from all putative viral contigs, using hmmscan with default values and an e-value cutoff of 0.001. However, the final taxonomic assignment provided for the putative viral contigs relies exclusively on our curated set of ViPhOGs.

To compare the annotations provided by all five databases, we ran hmmscan on all predicted proteins from putative viral contigs detected in the two mock community assemblies (Neto and Kleiner) and the 243 TARA Oceans assemblies. To allow for a better comparison against the other databases, we distinguished hits against the ViPhOG models into hits only based on an e-value cutoff of 0.001 and hits additionally identified using our predefined model-specific bit score thresholds.

### General implementation of VIRify

VIRify is implemented using the workflow management system Nextflow [30], and an overview of the pipeline is given in Fig 1. All third-party tools are encapsulated in software containers to allow easy distribution of the pipeline on local, cluster, or cloud systems. Custom Python scripts connect the output and input of the tools used and are available via GitHub (https://github.com/EBI-Metagenomics/emg-viral-pipeline/tree/master/bin). With configured Nextflow and Docker [68] installations, VIRify can be simply downloaded and run with a single command: nextflow run EBI-Metagenomics/emg-viral-pipeline -r v1.0—help. All required databases and metadata files are automatically downloaded and stored for later (offline) reuse. We always recommend running a stable release version from the repository, which can be selected via the -r flag. Different Nextflow profiles allow the execution on a local system or cluster (currently supported are LSF and SLURM). The pipeline may also be run in an "annotation" mode that skips the prediction of putative virus sequences and directly assigns viral taxonomies to all the input contigs. Please note that until version v1.0, the pipeline was implemented back-to-back in Nextflow and CWL [69]. Both workflow implementations were using the same scripts and software containers. However, to reduce maintenance and to fully focus on one implementation, from v1.0 all pipeline updates will be introduced only in Nextflow.

## Results

### Viral diversity is comprehensively covered by the set of manually curated informative ViPhOGs

The original ViPhOG database [24] consists of 31,150 profile HMMs that were created using proteins from viral genomes found in NCBI's databases. Based on the search of homologous

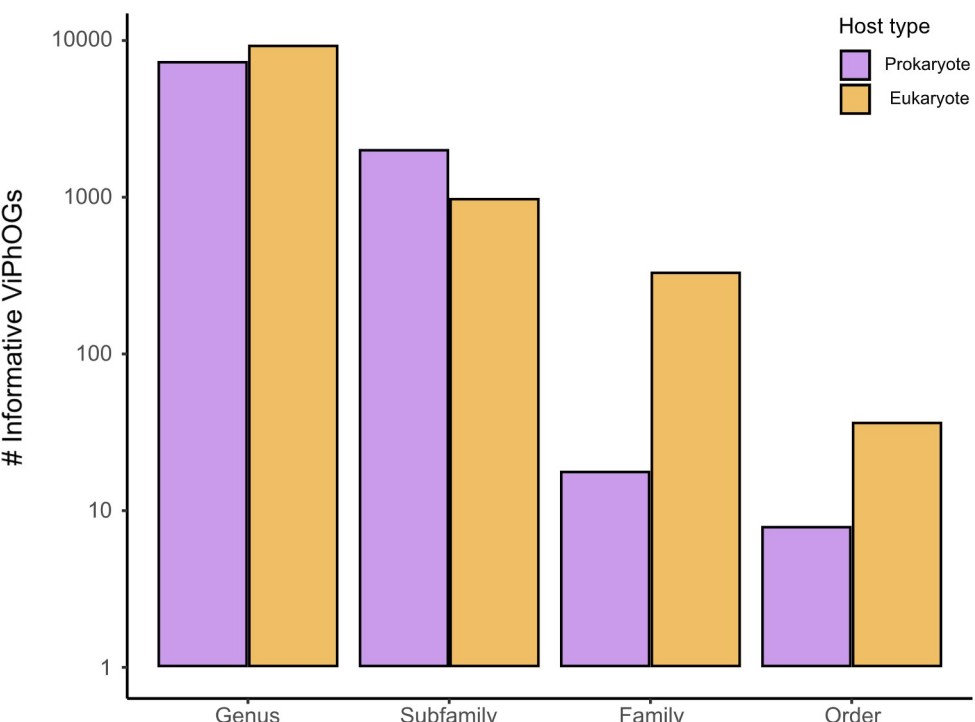

**Fig 2. Number of informative ViPhOGs identified for different viral taxonomic ranks.** 31,150 ViPhOGs were searched against all entries in UniProtKB, and based on the output they were designated as specific for different viral taxa (see Methods). Purple refers to specific ViPhOGs assigned to prokaryotic viral taxa, whereas yellow indicates specific ViPhOGs for eukaryotic viral taxa.

sequences in the UniProtKB database and the subsequent analysis conducted on the output data, we identified 20,266 informative ViPhOGs that could serve as taxonomic markers for viral taxa present in the current version of ICTV's taxonomy (MSL#38). **Fig 2** indicates the number of informative ViPhOGs obtained for each of the analysed taxonomic ranks (genus, subfamily, family and order), and for each rank, it illustrates the number of targeted taxa that correspond to either prokaryotic or eukaryotic viruses. Overall, 83.1% of the informative ViPhOGs were linked to taxa at the genus level and the numbers identified for the remaining ranks decreased from the subfamily to the order level. Furthermore, the percentage of informative ViPhOGs associated with eukaryotic viral taxa was slightly larger than the percentage of those associated with prokaryotic viral taxa (53.2% and 46.8%, respectively).

Regarding the coverage of currently known viral taxa, the informative ViPhOGs cover 18.6, 31.8, 20.9 and 10.8% of genera, subfamilies, families and orders represented in the viral NCBI taxonomy retrieved in January 2023 (**S1 Fig**). A comparison with the coverage of the viral NCBI taxonomy from March 2020 demonstrated an evident reduction in taxonomic coverage for all analysed ranks, which in turn provides a clear picture of the broad changes the viral taxonomy has undergone in the last few years. Nevertheless, our set of informative ViPhOGs includes taxonomy markers for many representative viral lineages within the ubiquitous and widely-described class *Caudoviricetes*, the eukaryotic virus order *Herpesvirales*, and the eukaryotic viral families *Mimiviridae*, *Coronaviridae*, and *Poxviridae*, among others. Moreover, based on the coverage of the NCBI viral taxonomy of January 2023, our set of informative ViPhOGs covers 38.6% of all the lineages represented by the currently known viral diversity in at least one taxonomic rank (**S1 Fig**).

## VIRify accurately classifies viruses on co-assemblies of two largely viral mock communities

VIRify was tested on datasets generated from viral mock communities to evaluate its viral sequence prediction performance and compare its taxonomic assignments with the taxonomy of the viruses present in the communities. The selected datasets were generated for two different studies that evaluated a range of viral enrichment and purification procedures that were applied to mock microbial communities. Kleiner et al. [32] sequenced a mock microbiome prepared in mouse faeces using bacterial and phage cultures (germ-free C57BL/6 J mice). The samples contained six different phages in varying concentrations: P22, T3, T7, ϕ6, M13 and ϕVPE25. Neto et al. [33] constructed a mock microbiome comprised of four common gut bacteria and nine highly diverse viruses, most of which represent the eukaryotic viral families *Circoviridae*, *Parvoviridae*, *Polyomaviridae*, *Alphaflexiviridae*, *Reoviridae*, *Coronaviridae*, *Herpesviridae* and *Mimiviridae*, and including a phage from the family *Ackermannviridae* (formerly classified in the family *Myoviridae*). The nine viruses differ highly in genome length (1.8 kb to 1,180 kb), genome type (dsDNA, dsRNA, ssDNA, ssRNA), and genome compositions (linear, circular, segmented). Co-assemblies of datasets obtained from viral-enriched samples were generated for each of the selected studies and used as input for the VIRify pipeline. The Kleiner co-assembly comprises 5,310 contigs, of which 224 are ≥ 1.5 kb with an N50 of 74,309 bp. The Neto co-assembly comprises 341,587 contigs, of which 1,686 are ≥ 1.5 kb with an N50 of 16,751 bp. Thus, 95.78% of the contigs in the Kleiner assembly, and 99.51% of the contigs in the Neto assembly are shorter than 1500 bp.

**VIRify comprehensively selects putative viral contigs.**   We ran a modified version of WtP (see Methods) on both mock community assemblies (Kleiner, Neto) to compare ten different tools for virus prediction. **S2 Fig** illustrates the performance of all the assessed tools for each of the analysed mock community assemblies. Our results show that a selection of VirSorter, VirFinder, and PPR-Meta sufficiently represented a common proportion of putative viral contigs from both mock community assemblies across all compared prediction tools (**Figs 3A**, **S2** and **S3**). Thus, the combination of these tools was selected as VIRify's method for viral contig prediction, although the selection of putative viral contigs from this combination is subjected to a set of rules similar to the selection criteria used in a previous study of global oceanic viral populations (see Methods section for further details) [12]. **Table 1** summarises the viral contig prediction results obtained with VIRify for the Kleiner and Neto assemblies.

For the Kleiner assembly, 17 contigs were identified as viral based on their alignment to the genome sequences of the phages that comprise the corresponding mock community (Fig 3**A**). VIRify identified 76.5% of the viral contigs in the Kleiner assembly and reported only one false positive prediction (NODE_192_length_1740_cov_2.233828), which also received no CheckV quality score. For the remaining contigs that VIRify predicted as viral, CheckV categorised six as high-quality, one as medium-quality, six as low-quality and the rest were undetermined (Fig 3A). All true negative predictions matched entries in NCBI's RefSeq corresponding to *Salmonella* genomes, an observation consistent with the significant amount of *Salmonella* DNA that was previously detected in all of the viral-enriched samples used for obtaining the Kleiner assembly [32]. Regarding the reported false negatives, 3 corresponded to prophages predicted in the genome of *Salmonella enterica* subsp. *enterica* serovar Typhimurium str. LT2 and the remaining one matched entries in NCBI related to phage M13. According to the calculated F1-scores, the viral prediction tools that performed best for this assembly were VirSorter (using the virome decontamination mode) and VIRify, both of which had an F1-score of 0.84 (**S2 Fig**).

Regarding the Neto assembly, alignment of contigs to the genome sequences of the mock community viruses identified 259 of them as viral. In this case, VIRify identified 51.7% of the viral

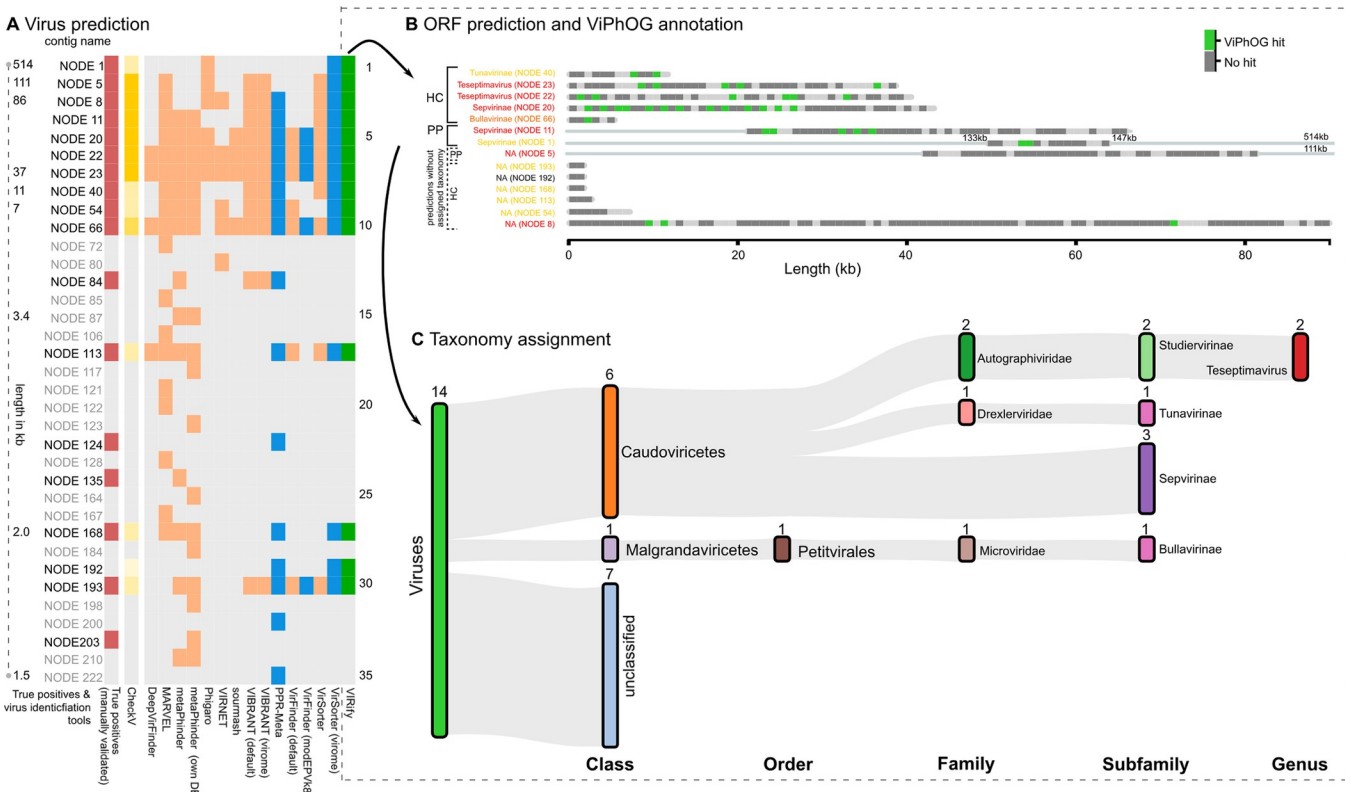

**Fig 3. Exemplary viral contig selection and annotation procedure for the Kleiner co-assembly. (A)** Comparison of virus predictions for the Kleiner co-assembly performed with various tools run via WtP. Shown are the 35 contigs (rows) predicted as viral by at least one of the tested tools. The column with red squares highlights the contigs manually identified as viral, as described in the methods section. Blue squares indicate contigs predicted as viral by VirSorter (virome decontamination mode), VirFinder (VF.modEPV_k8.rda model) or PPR-Meta. The column with green squares indicates the viral contigs reported by the prediction approach implemented in VIRify, based on the results from WtP (see Methods section). Yellow squares indicate CheckV-quality results for contigs selected by VIRify that are either high-quality, medium-quality, low-quality or not-determined; going from dark (high-quality) to light yellow (not-determined). **(B)** ORFs predicted with Prodigal and annotated with the informative ViPhOGs for the 14 contigs identified as viral by VIRify. Of these, eleven were predicted as high confidence (HC) and three as putative prophages (PP). No low-confidence viral predictions were reported for the Kleiner co-assembly. The coloured contig labels indicate the CheckV scores: red—high-quality, orange—medium-quality, yellow—low-quality, and black—not-determined. Dark grey bars indicate predicted ORFs without any ViPhOG hit, while green bars indicate ViPhOG hits based on the model-specific bitscores. **(C)** Predicted viral sequences and corresponding taxonomic assignments based on informative ViPhOG hits for the Kleiner co-assembly.

contigs present in the assembly, and 21.6% of the contigs reported as viral were false positives. Among the false negatives, 75.2% of the contigs were derived from the genome of the mimivirus included in the mock community. In addition, 23.2% of the false negatives were contigs that ranged in size from 1,515 to 5,906 bp and that corresponded to prophages identified in the genomes of the *E. coli* and *B. thetaiotaomicron* strains included in the mock community. Based on the F1-scores, the tools that performed best for the Neto assembly were PPR-Meta (0.84), VirFinder with the VF.modEPV_k8.rda model (0.63) and VIRify (0.62) (**S2 Fig**). In total, VIRify predicted 170 contigs of the Neto co-assembly as viral, of which CheckV rated two as high-quality, three as medium-quality, 142 as low-quality, and 24 as indeterminate.

**Table 1. VIRify's viral prediction results for two mock community assemblies.** The analyses were conducted for all assembled contigs longer than 1.5 kb.

| Assembly | Total contigs | Viral contigs | True positives | True negatives | False positives | False negatives |
|---|---|---|---|---|---|---|
| Kleiner | 224 | 17 | 13 | 206 | 1 | 4 |
| Neto | 1686 | 259 | 134 | 1390 | 37 | 125 |

Overall, the calculated F1-scores revealed that VIRify and PPR-Meta were the tools that performed best for predicting viral contigs from the analysed mock community assemblies (S2 Fig). VIRify had a better performance than PPR-Meta on the Kleiner assembly, particularly due to its ability to predict prophages from assembled bacterial contigs (e.g. NODE_1_length_514496_cov_72.252612 and NODE_5_length_111535_cov_40.991774) and due to PPR-Meta being less precise than VIRify, as evidenced by the higher number of false positive predictions reported by the former (Fig 3A). By contrast, PPR-Meta performed better than VIRify on the Neto assembly because the former tool had a higher recall rate that allowed the recovery of 87.6% of the mimivirus contigs, whereas VIRify recovered only 55.2% of the contigs derived from this virus.

**Taxonomic classification of phage contigs and putative prophage regions down to the subfamily and genus ranks.**   The taxonomic annotation performed with VIRIfy on the Kleiner assembly classified six of the putative viral contigs as members of the class *Caudoviricetes* (Fig 3B and 3C). Among them, two within the genus *Teseptimavirus*, three within the subfamily *Sepvirinae*, and one within the subfamily *Tunavirinae*. In addition, one contig corresponding to phage ΦX174, which is commonly used as a positive control in sequencing experiments, was correctly classified as a member of the subfamily *Bullavirinae* (class *Malgrandaviricetes*). Among the seven contigs with an assigned taxonomy, two were predicted by VirSorter to be putative prophage sequences (Fig 3B) which likely derived from bacterial contamination in the samples [32]. Of the six phages potentially included in the Kleiner co-assembly, VIRify identified and classified contigs that correspond to phages P22 (NODE_20), T3 (NODE_23), and T7 (NODE_22). While contigs derived from phages T3 and T7 were correctly classified in the *Teseptimavirus* genus, the contig from phage P22 was incorrectly classified as a member of subfamily *Sepvirinae*. However, both phage P22 and subfamily *Sepvirinae* belong to class *Caudoviricetes*, thus VIRify was able to classify the contig from phage P22 into the correct class. Overall, two of the phages from the original mock community present among the putative viral contigs were correctly classified by VIRify at the genus rank, one contig from phage ΦX174 was correctly classified at the subfamily rank, and one additional phage from the mock community was correctly classified at the class rank.

The pipeline reported a long 86 kb contig (NODE_8) from ϕVPE25 among the high-confidence viral contigs, whose taxonomic lineage was not identified, likely due to the low number of ViPhOG hits reported (Fig 3B). Phage M13 could not be identified either, as it was only assembled in small fragments due to low sequencing coverage, an observation that was consistent with the results reported previously [32]. The M13 genome is 6.4 kb long and metaSPAdes only recovered two contigs in the high confidence set that matched different parts of the phage's genome (NODE_113 with 2,682 bp and NODE_193 with 1,739 bp). Similarly, phage ϕ6 was not even detected by [32] after sequencing their viral-enriched samples, which explains why no contigs were identified for this phage in our coassembly.

**Comprehensive detection of prokaryotic and eukaryotic viruses from a highly diverse mock-virome.**   Taxonomic annotation of the reported 170 putative viral contigs in the Neto assembly revealed that 6 of them were classified as members of the class *Caudoviricetes* and 115 were classified in the orders *Imitervirales* (107 contigs), *Nidovirales* (2), *Ortervirales* (2), *Herpesvirales* (1), *Petitvirales* (1), *Pimascovirales* (1), and *Reovirales* (1) (Fig 4). Among them, we found members of the virus families *Ackermannviridae*, *Mimiviridae*, *Drexlerviridae*, *Herpesviridae*, *Microviridae*, *Iridoviridae*, *Sedoreoviridae*, *Coronaviridae*, and *Retroviridae*. For all these families, VIRify was also able to classify contigs at the subfamily level and some were also classified at the genus level: 97 contigs to *Mimivirus*, 2 contigs to *Alphacoronavirus*, 1 contig to *Limestonevirus*, and 1 contig to *Rotavirus* (Fig 4). The taxonomic classifications provided by VIRify were compared with the taxonomy of the genomes from which the analysed contigs

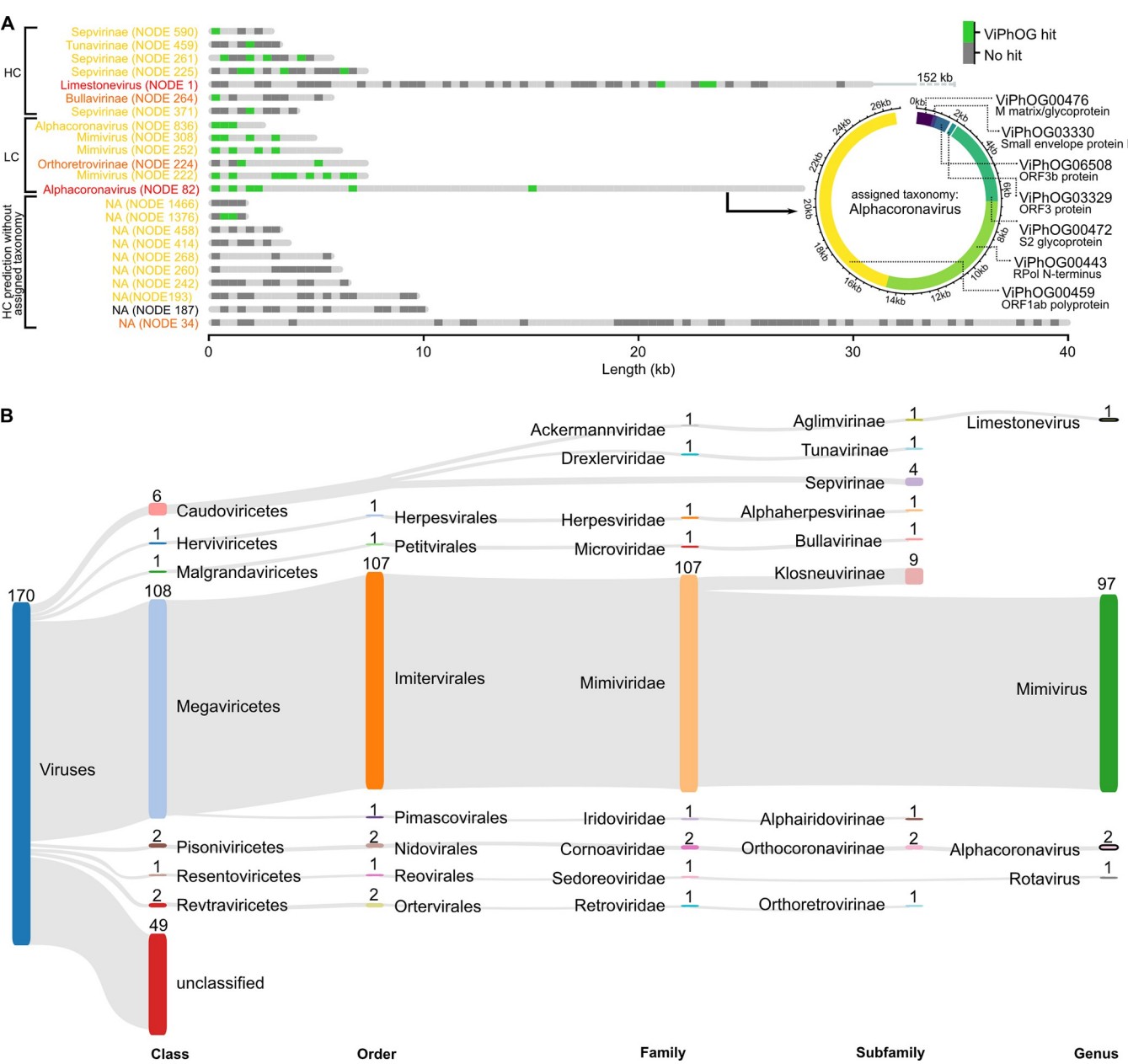

**Fig 4. Predicted ORFs and corresponding ViPhOG annotations and taxonomy assignments for the Neto co-assembly. (A)** ViPhOG-annotated ORFs for all contigs predicted as viral with high confidence (HC) and low confidence (LC) for the Neto assembly. Note that due to the high number of LC hits, only a selection of contigs is shown. The coloured contig labels indicate the CheckV scores: red—high-quality, orange—medium-quality, yellow—low-quality, and black—not-determined. VIRify assigned the genus *Alphacoronavirus* to NODE 82 based on seven informative ViPhOG model hits that are additionally shown as a circular visualization. NA—no taxonomy could be assigned due to missing model support. **(B)** Neto contigs predicted as viral with high and low confidence and their taxonomic assignments based on the ViPhOG model hits. Both visualizations can be automatically produced by VIRify and were only slightly manually adjusted via Inkscape.

were derived. We determined as a result that VIRify's taxonomic classification had an accuracy of 94.2% for the putative viral contigs detected in this assembly.

Contigs classified as members of subfamilies *Sepvirinae* and *Tunavirinae* matched entries in NCBI's nr database corresponding to phages of *E.coli*, suggesting that these were most likely derived from DNA contamination by the bacterial component of the mock community. No

contigs from the viruses that represented the families *Circoviridae*, *Polyomaviridae*, and *Alphaflexiviridae* were detected among the length-filtered contigs by blastn, and therefore no contigs were assigned to any of the corresponding taxonomic lineages. Eight contigs ranging from 1,543 to 2,356 bp and coming from viruses belonging to the families *Parvoviridae*, *Sedoreoviridae* and *Herpesviridae* were detected among the set of low-confidence viral contigs. One of these was correctly classified at the subfamily rank (*Alphaherpesvirinae*) and another was correctly classified at the genus rank (*Rotavirus*). No contig was classified within the viral lineage that includes the family *Parvoviridae*.

One 7,051 bp putative viral contig was classified in the subfamily *Orthoretrovirinae*, which does not match the lineage of any of the viruses included in the mock community. However, when this contig was used as query for a search of NCBI's nr nucleotide database with megablast, the top hits corresponded to porcine endogenous retroviral sequences, including several complete genomes of this type of virus (e.g. accession number EF133960.1, 97.93% identity, e-value 0.0). This result indicates that the classification of the mentioned contig as a member of *Orthoretrovirinae* was correct, and that this virus was a contaminant that was likely present in the stock of the Porcine circovirus included in the mock community.

## VIRify comprehensively reveals virus taxonomies in global ocean ecosystems

To showcase the utility of VIRify in uncovering viral diversity in ocean environments, we ran VIRify on 243 assemblies provided by the TARA Oceans project [12,17,18], including 20 samples processed via 0.1–0.22 μm filters and thus potentially enriched for viral sequences [12]. As expected, we identified more complete/high-quality viruses in such samples enriched for smaller viruses in comparison to samples derived from larger 0.45–0.8 μm filters (**Fig 5A**). Across all assemblies and including high and low confidence viral predictions, the following viral families were detected with at least 10 putative viral sequences assigned to them: *Straboviridae* (n = 4,128), *Mimiviridae* (3,627), *Phycodnaviridae* (2,281), *Drexlerviridae* (931), *Herelleviridae* (764), *Salasmaviridae* (363), *Autographiviridae* (326), *Microviridae* (83), *Poxviridae* (37), and *Iridoviridae* (31) (**S4 Fig**). In addition, we saw differences in the amount of predicted viral contigs between samples obtained from the same location but with DNA extracted using different filter sizes. For example, in CEUO01 (TARA_124_MIX) filtered for 0.1–0.22 μm (SAMEA2622799) 219 contigs were assigned to the class *Caudoviricetes*, whereas for the same sample material, but filtered for 0.45–0.8 μm (SAMEA2622801), only 32 *Caudoviricetes* contigs were found (Fig 5A). Interestingly, members of the genus *Prasinovirus* (large double-stranded DNA viruses that belong to the order *Algavirales*) were predominantly found in low confidence sets of 0.22–3 μm filtered fractions, which suggests that these viruses would have been missed if only VirSorter had been used on the data (Fig 5B).

A previous study in which the aforementioned marine samples were analysed reported that the majority of identified viruses were members of the recently abolished viral families *Myoviridae*, *Siphoviridae* and *Podoviridae* [12,70]. Following these changes in the viral taxonomy, we can assume that the majority of the viruses identified in the mentioned study were members of the class *Caudoviricetes*, as this new taxon groups all tailed viruses formerly classified in the three abolished families. The results obtained with VIRify confirmed this observation, as 52% of the contigs classified at the family rank were assigned to taxa within the class *Caudoviricetes*. Furthermore, the family *Phycodnaviridae* had been identified as one of the most abundant viral families in the marine samples, and the results obtained with VIRify confirmed this observation.

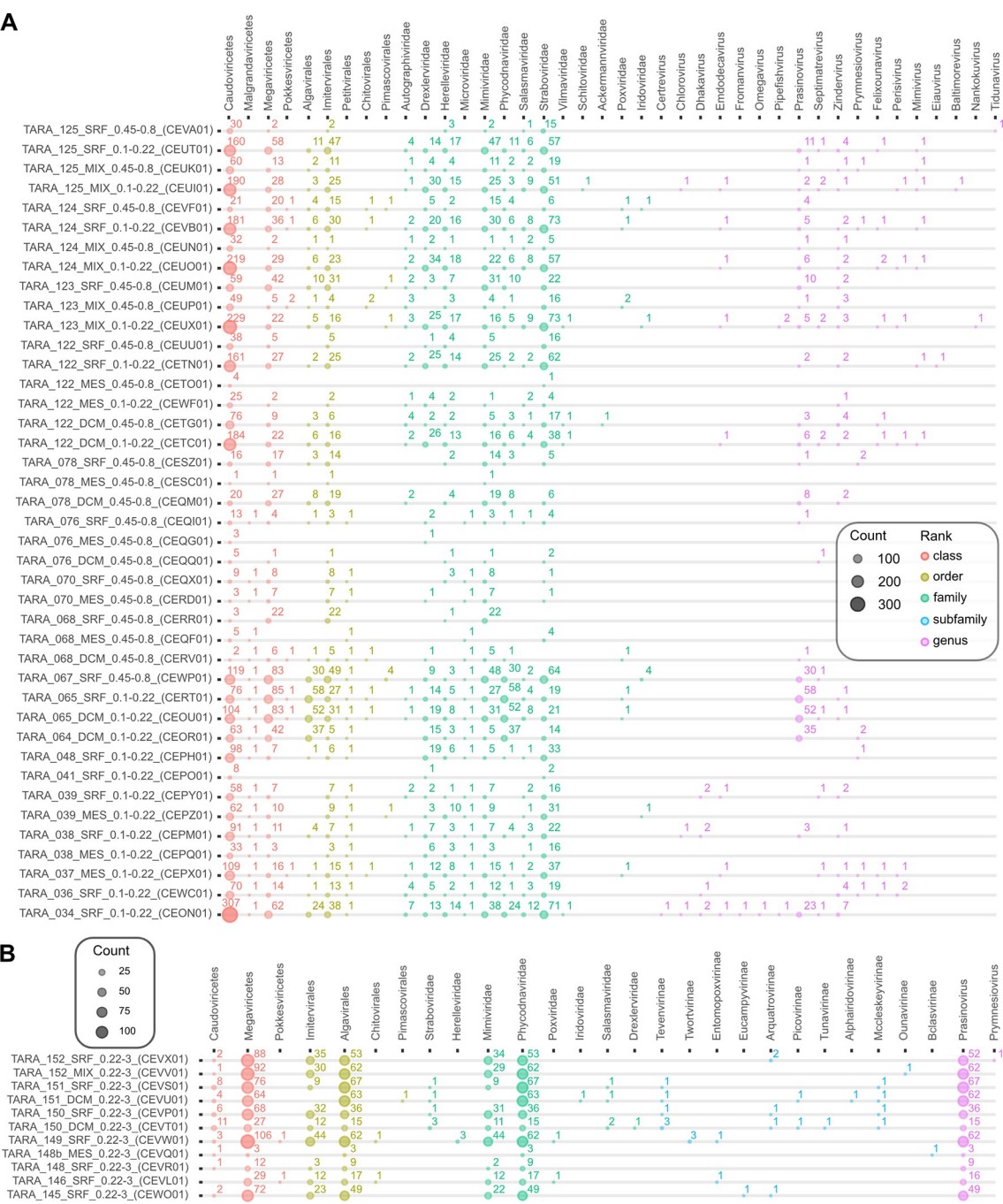

**Fig 5. Viruses predicted and annotated by VIRify for 243 TARA Oceans assemblies.** The assembly identifiers include information about the size fractionation of the corresponding sample. For example, samples obtained by smaller filter size 0.1–0.22 μm (and so expected to be enriched for smaller viruses) are labelled with the suffix _0.1–0.22. **(A)** Shows a selection (filters 0.1–0.22 μm and 0.45–0.8 μm) of 41 samples and the number of predicted viruses based on high confidence (VirSorter categories 1 and 2) and low confidence (VirSorter category 3 and combined VirFinder and PPR-Meta results) hits. More viruses are found for smaller filter sizes, as expected. Assemblies based on smaller filter sizes are highlighted in bold. For visualization purposes, we summarize high and low confidence predictions for samples labelled with the filter sizes 0.1–0.22 and 0.45–0.8. **(B)** Selection of 0.22–3 μm filtered samples with a high number of predicted prasinoviruses, large double-stranded DNA viruses belonging to the order *Algavirales*. These viruses are predominantly found in the low confidence set; thus they would have been missed if only VirSorter were run on the data but are predicted by our combination of VirFinder and PPR-Meta.

### VIRify expanded the number of taxonomically classified VCs from the GPD and provided higher resolution classifications

Genetically-related entries in the GPD had been previously sorted into 57,866 VCs, using a graph-based clustering approach [34]. Using the reported CheckV quality and completion values, we selected a representative sequence from each VC. According to the published GPD metadata, 7,432 of these representative sequences had been previously classified and 99% of them were reported as members of the abolished phage families *Myoviridae*, *Siphoviridae* and *Podoviridae*. We analysed the selected representative sequences with VIRify to assess whether our pipeline could provide updated taxonomic classifications for them.

VIRify provided taxonomic classifications for 13,651 of the analysed representative sequences, which corresponds to an increase of 83.7% from the number reported previously. The vast majority of these classifications were to lineages within the class *Caudoviricetes* (98%), whereas most of the remaining sequences were classified in the families *Iridoviridae*, *Microviridae* and *Herpesviridae* (**Fig 6**).

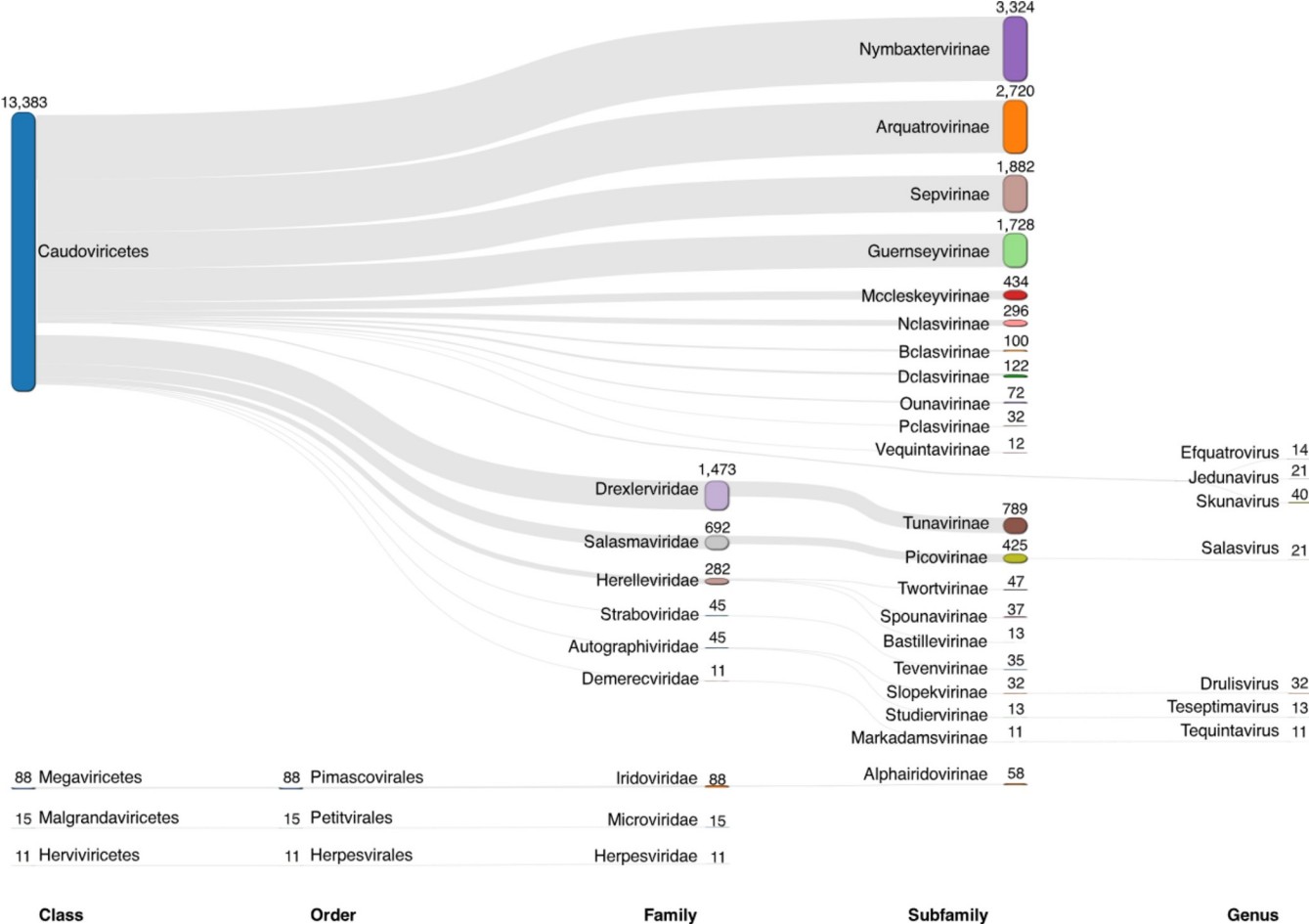

**Fig 6. Taxonomic classification of representative sequences from the GPD.** One representative sequence from each of the 57,866 VCs present in the GPD was selected based on their reported CheckV quality and completion values. The representative sequences were analysed with VIRify using the '—onlyannotate' option to obtain taxonomic classifications for them. The taxa displayed in the Sankey plot correspond to those in which at least 10 representative sequences were classified.

The families and subfamilies most commonly reported in the results include phages that target bacteria from the phyla Proteobacteria, Bacillota (i.e. Firmicutes) and Actinomycetota, all of which have been reported as common members of the human gut microbiome [71]. In particular, the families *Drexlerviridae*, *Salasmaviridae* and *Herelleviridae* have already been identified as frequent members of the human gut virome [72,73]. Interestingly, the sequences classified in the subfamilies *Nymbaxtervirinae* and *Arquatrovirinae* corresponded to putative phages targeting bacteria from either the Bacillota or Actinomycetota phyla. Previous studies have provided evidence indicating that phages targeting bacteria from these two phyla share a common ancestry, which hints at the possibility of cross-phyla phage-host interactions [74].

## ViPhOG models comprehensively cover viral proteins

Our comparison of virus-specific protein profile HMM databases included in VIRify showed that the ViPhOG models covered a large proportion of potentially viral CDS in agreement with the other databases (Fig 7A). In total, we predicted 1,204,455 CDSs from all high-confidence viral contigs that VIRify reported for the 243 TARA assemblies, of which the ViPhOGs covered 56.7% (49.4% with applied bit score threshold) and thus were only slightly outperformed by the VOGdb (56.8%). The largest set of shared annotations comprised 284,791 CDS annotated as viral based on models from all compared databases except the RVDB (Fig 7A). In addition, the RVDB had the least predictions, with only 15.3% annotated CDS compared to all other databases. Our model-specific bit score filtering reduces the amount of CDS annotations derived from ViPhOGs by 7.3% (87,869 CDS) (Fig 7A). While this might lead to the loss of potentially informative annotations that could be used for taxonomy assignment, we also remove false positive model hits, as shown in Fig 7B. Interestingly, the VPF derived from the IMG/VR database, which also includes novel viral sequences derived from metagenome approaches, comprises a large proportion of unique models that match 117,962 CDS (9.8%) that are not covered by any of the other annotated databases. On the other hand, a significant number of CDS are annotated by models from the other databases but missed by the VPF models.

## Discussion

Metagenomic surveys of different environments during the last few decades have had a profound impact on the rate at which novel viruses are discovered [75]. Despite the resulting sharp increase in the number of publicly available viral genomes, viruses remain relatively understudied in many environments and the taxonomic profiling of viral communities is still challenging due to the lack of universal genetic markers that support the phylogenetic resolution of viral taxa [76,77]. As a contribution to the currently available repertoire of tools designed to address these challenges, we designed and implemented the VIRify pipeline for detecting and taxonomically annotating viral contigs in metagenomic datasets.

The results presented here for the mock community assemblies demonstrated that VIRify is a suitable pipeline for generating highly accurate taxonomic profiling of viral communities present in metagenomic datasets. Compared with most of the tools currently used for detecting viral contigs in metagenomic assemblies, VIRify demonstrated higher predictive performance for contigs ≥ 1.5 kb in both the Keiner and Neto assemblies (**Figs 3A, S2** and **S3**). Furthermore, using our manually-curated informative ViPhOGs led to the taxonomic classification of putative viral contigs from the Neto assembly with an accuracy of 94.2%. Regarding the Kleiner assembly, all classified putative viral contigs were assigned to the correct taxonomic lineages at different degrees of resolution, with some classified up to the family rank and others

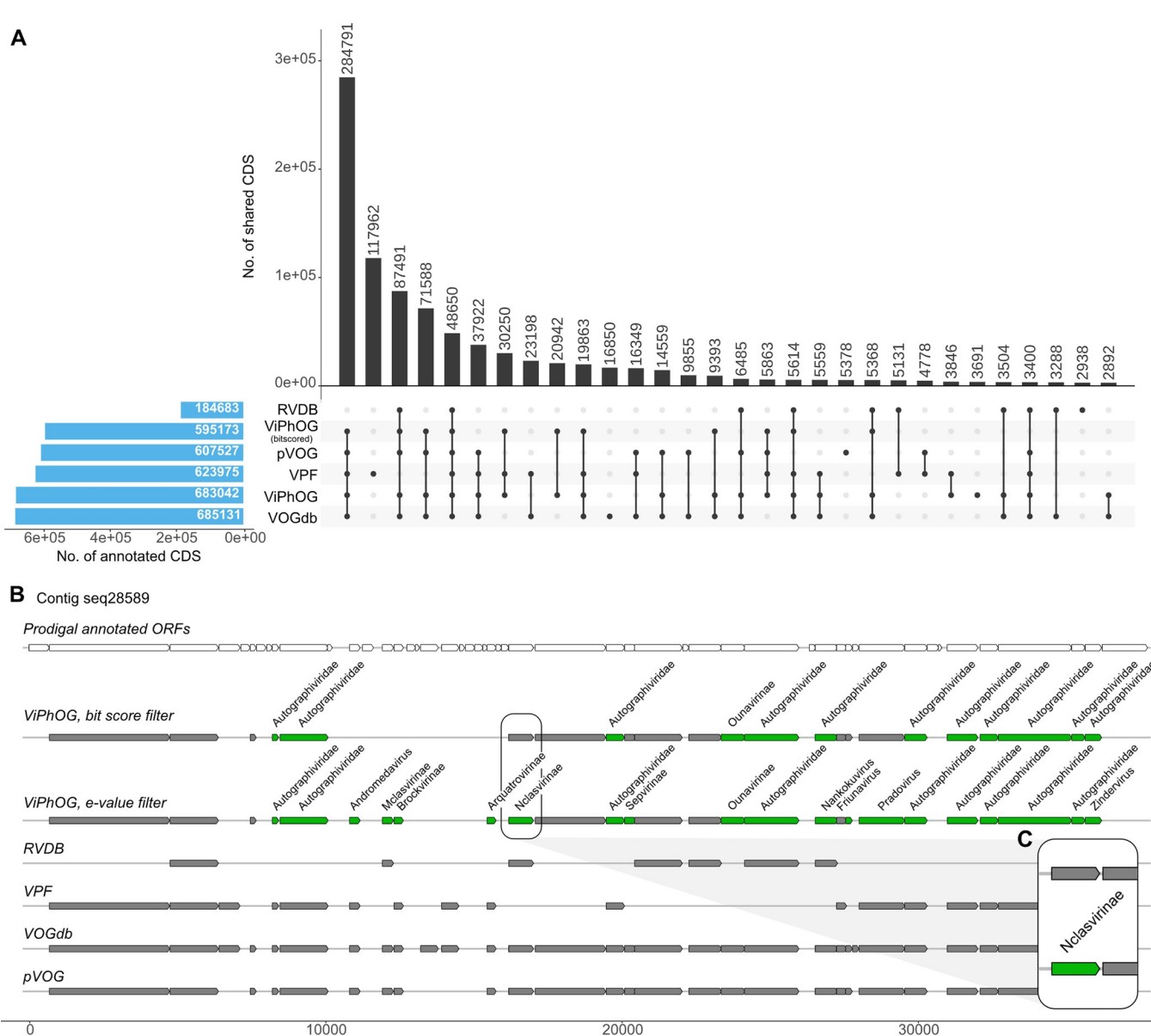

**Fig 7. (A)** Comparison of annotated CDS from VIRify's high-confidence viral predictions from all 243 TARA Oceans assemblies. Our comparison shows that the ViPhOG models comprehensively cover a large proportion of potentially viral CDS in agreement with other public databases. The RVDB had the fewest predictions, with only 15.3% annotated CDS. Results of the VPF database, where models are derived from the IMG/VR database that also includes novel viral sequences derived from metagenome approaches, comprise a large proportion of unique models that exclusively match 117,962 CDS (9.8%) that are not covered by any of the other databases. However, a significant number of CDS are annotated by models from the other databases but missed by VPF. Our model-specific bit score filtering (ViPhOG—threshold) reduces the number of CDS annotations derived from ViPhOGs by 7.3% **(B)** Visualization of predicted (top row) and annotated ORFs for one exemplar contig from TARA Oceans assembly CEUI01. Grey bars indicate hits against an HMM of the corresponding database, while informative ViPhOG hits with taxonomic information are shown in green. The top ViPhOG track shows hits filtered by bit score (or e-value if no bit score could be assigned for a model, see Methods) and the bottom ViPhOG track shows e-value-filtered hits. **(C)** While the model-specific ViPhOG bit score threshold can lead to the loss of potentially informative annotations for taxonomy assignment, it can also reduce the number of false positive model hits and thus increase the overall accuracy of VIRify.

classified up to the genus rank. The pipeline also provided taxonomic classifications for putative prophages identified in contaminant bacterial contigs from both mock assemblies.

VIRify's viral prediction performance was the result of combining the effectiveness of Vir-Sorter in predicting phage and prophage sequences [51,78], with the ability of PPR-Meta and

VirFinder's VF.modEPV_k8.rda model to predict sequences from eukaryotic viruses, as evidenced by their F1-scores for the Neto assembly (**S2 Fig**). Overall, most of the viral prediction tools evaluated here demonstrated a low performance for the Neto assembly, which indicates that these tools were mainly developed for the detection of phages and are not suited for the comprehensive detection of eukaryotic viral sequences. Considering that PPR-Meta was originally designed for detecting phages and plasmids [56] and that the Neto assembly mainly comprised eukaryotic viruses, it was striking to find that PPR-Meta was the tool that performed best for this assembly. Nonetheless, we decided to keep the combination of VirSorter, VirFinder and PPR-Meta as VIRify's viral prediction approach for two main reasons. The first is that PPR-Meta's underlying model was not explicitly trained to detect prophages in bacterial genomic sequences. As the detection of both free viruses and prophages was critical to the aim of providing comprehensive viral community profiling, we needed to include another viral detection tool capable of efficiently predicting prophages. VirSorter provides this functionality, as has been demonstrated by many previous studies in which it was employed to detect prophages in genome sequences from a diverse range of bacteria [13,78]. The second reason for not using PPR-Meta on its own is that, despite its high sensitivity, it tends to call a relatively higher number of false positives in comparison to other viral predictors [79]. Considering PPR-Meta's performance on the mock community assemblies, we reasoned it would be advantageous to keep this tool within VIRify's arsenal by combining its predictions with the ones obtained with VirFinder to limit the number of false positive predictions.

Several studies have reported collections of profile HMMs for groups of orthologous viral genes, such as pVOGs [67], vFAMs [80], RVDB [66] and VOGdb (http://vogdb.org/). Even though some of these have been employed for viral taxonomic profiling, our informative ViPhOGs stand out due to their bespoke bit score thresholds that were carefully selected to decrease the likelihood of false positive taxonomic assignments. As illustrated in Fig 7B, using the informative ViPhOGs with a single e-value threshold resulted in a higher chance of false positive taxonomic annotations, as was the case for contig seq28589 from the assembly of the TARA Oceans sample CEUI01 (study accession PRJEB7988). Therefore, our results indicate that using curated bit score thresholds offers an adequate balance between predictive power and classification accuracy. This strategy has been successfully implemented previously in Pfam and TIGRFAMs, some of the most widely used protein family databases for the functional annotation of genomes and metagenomes [35,81].

Another novelty of our curated set of informative ViPhOGs is the use of TSR to set taxonomic assignment thresholds specific for each taxon. The TSR measures the density of assigned informative ViPhOGs for each taxon present in the viral lineages covered by our set of models. Thus, setting the taxonomic assignment threshold with this parameter allows the algorithm to adjust its level of stringency for each individual taxon. This feature is particularly advantageous for taxa characterised by relatively small genomes, for which there is a fewer number of informative ViPhOGs in our collection. This benefit was especially evident during the analysis of the Neto assembly, where the use of this feature increased the number of classified contigs from several of the eukaryotic viruses present in the mock community. A run of VIRify with the taxonomy assignment threshold set to 0.6 for all taxa resulted in the absence of short contigs (1,543–2,356 bp) classified in the subfamily *Alphaherpesvirinae* and the genus *Rotavirus*. This result indicates that the use of taxon-specific thresholds improves the detection of short contigs, by lowering the taxonomic assignment stringency.

We further showed that VIRify can be applied to metagenomic assemblies obtained from large environmental studies, such as the TARA Oceans expedition [12,17,18]. In accordance with a previous report, a high proportion of predicted viruses could not be taxonomically classified into a known viral family [12]. However, the majority of contigs that VIRify classified

were assigned to different families within the class *Caudoviricetes*, which agrees with the previous observation that most of the classified viruses in these samples were members of the former phage families *Myoviridae*, *Siphoviridae* or *Podoviridae*. Also in agreement with the previously reported study, the family *Phycodnaviridae* was one of the taxa most commonly identified by VIRify among the analysed marine samples. Interestingly, prasinoviruses were predominantly found in low confidence sets, thus, would have been missed by only running VirSorter on the data (**Fig 5B**). Here, our combination of VirFinder and PPR-Meta predictions helped to recover contigs that the pipeline could later taxonomically assign to this genus. Due to their larger size, prasinoviruses are underrepresented in smaller size fractionated samples, underlying the importance of appropriate filtering and enrichment steps and their combination to comprehensively collect viruses from environmental samples.

The use of VIRify to analyse a set of representative sequences from the GPD resulted in a large increase in the number of taxonomically classified sequences (83.7%). The vast majority of the sequences were assigned to families and subfamilies within the class *Caudoviricetes*, which agrees with the former classifications reported in the GPD metadata. Furthermore, phages classified in the reported taxa have been previously found to target bacteria within some of the most frequently identified bacterial phyla in the human gut [71–73]. However, a few of the representative sequences were classified in the families *Iridoviridae* and *Herpesviridae*. Although the latter taxon has been reported as a frequent member of the human gut virome, it is likely that these results correspond to false positive assignments considering the phage sequence prediction approach followed in the generation of the GPD [34,82]. Nonetheless, the number of sequences assigned to these taxa corresponds to only 0.72% of the total number of representative sequences that VIRify classified. Overall, VIRify provided a more comprehensive and updated taxonomic characterisation of the viral sequences that comprise the GPD.

Despite the great performance observed for the analysed assemblies, our results revealed a few limitations of the current VIRify pipeline. As evidenced by the analysis of the Neto assembly, VIRify's ability to detect contigs from eukaryotic viruses is currently suboptimal. Increasing the pipeline's sensitivity will require the careful evaluation of novel viral prediction tools that could easily be incorporated or used to replace the ones currently used. For example, VirSorter might be replaced by the recently released VirSorter2 [83] in a future version of VIRify. An additional limitation of VIRify is the existing bias in the extent to which the informative ViPhOGs represent the different genera in the current viral taxonomy. We determined that our current collection of informative ViPhOGs covers 38.6% of the lineages that comprise the current viral taxonomy, which has evidently decreased in comparison with the value calculated for the NCBI viral taxonomy from March 2020 (89.3%). This reduction in coverage reflects the substantial changes that the viral taxonomy has experienced since the ViPhOGs were generated. Our results demonstrate that the informative ViPhOGs are a powerful resource for the taxonomic classification of contigs from covered viral taxa, but it is imperative to generate new ViPhOGs that expand the taxonomic coverage to provide the community with a general-purpose viral analysis tool. This task will be central to the upcoming developments of VIRify. Finally, the quality of the metagenomic assembly is key for VIRify's performance due to the challenge that short viral contigs pose on their detection by the viral prediction tools currently used in the pipeline (**Fig 3A**). Furthermore, short contigs will generally contain fewer complete CDS, and thus will be less likely to have the minimum number of ViPhOG hits required by the pipeline to provide a taxonomic assignment.

VIRify is a novel HMM-based resource for viral taxonomic classification that can be broadly classified as a protein similarity-based method. There are other types of approaches that have been efficiently implemented for viral taxonomic profiling, such as the use of kmer

matching to a reference database of complete phage genomes. Phanta, a recently developed tool for phage-inclusive gut microbiome profiling, takes advantage of a much larger number of phages discovered through extensive analyses of the human gut microbiome, in comparison with other biomes [84]. This tool uses a large database of gut phages and their associated taxonomic information to perform taxonomic profiling using kmer-based searches, which allows the analysis of phage communities from metagenomic reads. This is an advantage over methods that rely on metagenomic assemblies (such as VIRify) because the analysis of reads allows the detection of low-abundance phages that are difficult to assemble into contigs [85,86]. However, the advantage of using protein profile HMMs is that they gather viral reference data in a more efficient and compact way, in comparison with databases of complete genomes generally required by kmer-based methods. This is particularly relevant in the case of Phanta, which was designed to perform taxonomic profiling at the species level. Therefore, the combination of VIRify with a larger collection of ViPhOGs covering the entire viral taxonomy would be easier to implement as a general-purpose viral taxonomic profiling, than methods relying on the use of collections of complete reference genomes.

Although VIRify has been benchmarked and validated with metagenomic data in mind, it is also possible to use the pipeline to detect RNA viruses in metatranscriptome assemblies (e.g. SARS-CoV-2, see [87]). Some additional considerations in this regard include 1) quality control, 2) assembly, 3) post-processing, and 4) classification. Some brief recommendations on how to prepare metatranscriptome assemblies and run them through VIRify can be found in the GitHub manual.

Overall, our results demonstrated the utility of VIRify for analysing viral communities in metagenomic datasets. In contrast with other resources currently available, VIRify offers an enhanced capability to predict contigs from eukaryotic viruses and the ability to taxonomically classify viral contigs using a set of carefully curated HMMs and their bespoke bitscore thresholds. Future versions of VIRify will attempt to improve even further the sensitivity of viral contig prediction and to extend the ViPhOGs' taxonomic coverage by including sequences from uncultured viruses. In addition, the taxonomic classification of contigs from small viruses could be improved by adjusting the taxonomic voting system to accommodate differences in the number of available informative ViPhOGs and in the average genome size observed between different viral lineages. VIRify will be fully integrated into the MGnify suite of metagenomic analyses hosted at the EMBL-EBI, which is continuously supported and updated to suit the needs of the scientific community.

## Runtime and resources

The pipeline runtime can vary significantly depending on the hardware and data size. For optimal performance, we recommend running VIRify on high-performance computing (HPC) systems or the cloud, especially for larger datasets. On a typical laptop with 8 cores, the pipeline runs reasonably fast for small datasets, such as the two mock community data sets, taking approximately 16 minutes for the Kleiner co-assembly and 30 minutes for the Neto co-assembly. However, for larger datasets like the 243 TARA Ocean assemblies, the pipeline requires 2 days and 13 hours on an HPC system with SLURM and the default configuration profile for cluster execution. This runtime estimate includes the execution time for each process, excluding database downloads. Keep in mind that the actual runtime can be influenced by factors such as fluctuation and pending jobs due to high demand on the HPC. The RAM requirements of the VIRify processes are generally low and can be run on a decent laptop (minimum 8 GB). Experienced users can adjust CPU and RAM resources in the Nextflow configuration to optimize performance on an HPC or cloud infrastructure. The disk space required by VIRify is

generally reasonable compared to the input size. While Kleiner and Neto's output folders comprised 130 and 270 MB, respectively, the full TARA run (including virus prediction and taxonomy assignment) required 174 GB. The GPD run, which skipped virus prediction and performed only CDS detection and taxonomy assignment, required 12 GB of storage. In addition, intermediate files in the Nextflow working directory can take up much space; for example, 263 GB of intermediate data were written to the work directory for all 243 TARA assemblies, while less than 1 GB of space was used for Kleiner and Neto.

## Supporting information

**S1 Fig. Informative ViPhOGs' coverage of viral taxonomy.** Circular dendrograms showing the NCBI virus taxonomy from March 2020 (upper panel) and January 2023 (lower pannel), with taxa covered by informative ViPhOGs highlighted with either dots or bars, and using the same colour scheme as in **Fig 2**. For the taxonomic ranks (order, family and subfamily) dots were used to indicate whether any informative ViPhOGs were identified for the corresponding taxon. By contrast, bars were used for taxa in the genus rank (tree's leaves) to indicate the number of different informative ViPhOGs identified for the corresponding genus, as indicated by the numeric scale in the outer rings.
(PNG)

**S2 Fig. F-score comparison for virus prediction results from the Kleiner and Neto assemblies for all tools run via WtP and VIRify (combination of VirSorter, VirFinder with VF. modEPV_k8.rda model and PPR-Meta).**
(PNG)

**S3 Fig. UpSet output for the Neto assembly calculated with WtP.** Viral prediction tools are compared and overlapping sets are shown.
(PNG)

**S4 Fig. Abundance plot of viral ranks predicted for 243 TARA Oceans assemblies (https://www.ebi.ac.uk/ena/browser/view/PRJEB22493) using the VIRify pipeline.** Combined results for high confidence, low confidence, and putative prophage hits are shown. VIRify was run in v0.2 and with the—virome option and a contig length filtering of 5000 nt.
(PDF)

## Acknowledgments

The authors thank Franziska Hufsky for improving the readability of the illustrations. The authors would also like to acknowledge Lorna Richardson for contributing to the edition of the final manuscript version.

## Author Contributions

**Conceptualization:** Guillermo Rangel-Pineros, Alejandro Reyes Muñoz, Robert D. Finn.

**Formal analysis:** Guillermo Rangel-Pineros, Martin Hölzer.

**Investigation:** Guillermo Rangel-Pineros, Alexandre Almeida, Martin Hölzer, Robert D. Finn.

**Methodology:** Guillermo Rangel-Pineros, Alexandre Almeida, Alejandro Reyes Muñoz, Martin Hölzer, Robert D. Finn.

**Project administration:** Alejandro Reyes Muñoz, Robert D. Finn.

**Software:** Guillermo Rangel-Pineros, Martin Beracochea, Ekaterina Sakharova, Martin Hölzer.

**Supervision:** Alejandro Reyes Muñoz, Robert D. Finn.

**Validation:** Guillermo Rangel-Pineros, Martin Beracochea, Ekaterina Sakharova, Manja Marz, Martin Hölzer.

**Visualization:** Guillermo Rangel-Pineros, Manja Marz, Martin Hölzer.

**Writing – original draft:** Guillermo Rangel-Pineros, Martin Hölzer.

**Writing – review & editing:** Guillermo Rangel-Pineros, Alexandre Almeida, Martin Beracochea, Ekaterina Sakharova, Manja Marz, Alejandro Reyes Muñoz, Robert D. Finn.

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
