## [Decision Letter · Decision Letter 0]

10 Jan 2023

Dear Dr Rangel-Piñeros,

Thank you very much for submitting your manuscript "VIRify: an integrated detection, annotation and taxonomic classification pipeline using virus-specific protein profile hidden Markov models" for consideration at PLOS Computational Biology.

As with all papers reviewed by the journal, your manuscript was reviewed by members of the editorial board and by several independent reviewers. In light of the reviews (below this email), we would like to invite the resubmission of a significantly-revised version that takes into account the reviewers' comments, esp those by Reviewer 3.

We cannot make any decision about publication until we have seen the revised manuscript and your response to the reviewers' comments. Your revised manuscript is also likely to be sent to reviewers for further evaluation.

Sincerely,

Christos A. Ouzounis

Academic Editor

PLOS Computational Biology

Thomas Leitner

Section Editor

PLOS Computational Biology

Lucy Houghton

Staff

PLOS Computational Biology

Reviewer's Responses to Questions

**Comments to the Authors:**

Reviewer #1: The authors have addressed a large issue within the field, although issues remain with annotation, VIRify clearly competes well with other tools in the field.

The fact that the authors have used two mock communities is great, although I do find the fact that comintamination within them is noted as a potential issue, troubling. Clearly better mocks are required in future studies, but I believe that to beyond the scope of this paper.

Major issues;

- The basis for the HMMs is the viral genomes from 2015, this is 7 years ago, meaning alot of diversity is missing. Is this something that can be easily updated? If not, this poses a major limitation which the authors should address in the text of the paper.

- I see no justification for the 10% ViPhOG annotation cutoff or that atleast 2 hits are reported at a set level with 60% consistency. Please benchmark these factors to define why these thresholds are suitable.

Reviewer #2: Rangel-Pineros et al are describing a pipeline "VIRify" that can detect viral contigs in a metagenomic assembly and perform taxonomy annotations using protein-similarity. The pipeline takes a pre-assembled set of contigs as an input and classifies them using a combination of several viral-detection software tools. Contigs classified as viral are further subjected to the taxonomy characterization: protein-coding ORFs are predicted for each contig and these ORFs are matched against a set of protein domain models (informative ViPhOGs) representative for different viral genera, families or orders. Authors validate their choice of software packages for each step of the pipeline and then benchmark it against 2 metagenomic datasets with experimentally known combinations of viruses/phages. Selection of protein-domain models ( informative ViPhOGs) is briefly described in the manuscript but has been previously published elsewhere.

From the software perspective, "VIRify" adheres to the best practices: the pipeline is implemented using 2(!) Workflow management systems (nextflow and CWL), software packages VIRIfy relies on are containerized using docker/singularity for portability, documentations is thorough and contributors appear active on the github page of the repository.

The manuscript itself is well written and easy to follow for the most part, although it could be shortened somewhat by skipping redundant descriptions and unnecessary details in the introduction (see below).

I do have several comments that could help improve the manuscript:

Major:

1. Elaborate more on the significance of the “TARA Oceans'' analysis - right now it is unclear if ViRIfy provides novel insight into those datasets or if ViRIfy is valuable as a convenient tool/utility here ? Can “Massive expansion of human gut bacteriophage diversity” paper by some of the co-authors be used to strengthen the utility of VIRify’s approach ?

2. Reading the “Selection and comparison of virus prediction tools” section in Methods 239-286, the choice of 3 viral detection software seems somewhat arbitrary - i.e. according to SFig2 PPR-Meta outperformed most of the other tools - why not use it alone ? Figure 3 helps answer that a bit, but then maybe there is another way to justify the choice of VirSorter, VirFinder, PPR-Meta ?

Minor:

1. Shortening of the manuscript would improve readability, here are several examples:

a. 84-90 lytics vs lysogenic life cycles of phages - interesting - but in my opinion distracting

b. Shorten ViPhOG related description that were covered in the previous publication - e.g. in the first section of “Results”

2.Provide references to supplementary figures in “Selection and comparison of virus prediction tools” section

3.It would be nice to include a paragraph on ViRify’s runtime expectations - i.e. how long it would run on a “typical” dataset, what hardware is recommended

4.Several of the tools that VIRify depends on are deep learning based - do they require GPUs or CPU is sufficient? Do you take it into account for containerization ?

5. Could you discuss an alternative approach of taxonomy classification based on “complete” viral genomes (e.g. as in “Phanta: Phage-inclusive profiling of human gut metagenomes” preprint by Pinto et al) - what are some advantages/limitations relative to protein-similarity based approach ?

Reviewer #3: The manuscript by Rangel-Pineros and colleagues describes the creation of a new pipeline for virome analyses called VIRify. This pipeline is avalaible from GitHub as a Nextflow pipeline or CWL implementation. In brief, the pipeline uses either reads or assemblies, predicts which contigs are viral and provides a taxonomic classification using virus-specific protein profile HMMs derived from a separate study. The pipeline was tested on two virome mock communities and a TARA Oceans metagenomics study. The development included an analysis of the appropriateness of several different virus prediction tools using the “What the Phage” workflow (Marquet et al, 2022).

VIRify has a great strength that makes it stand out from other pipelines, the use of the ViPhOGs, a well-curated set of 22,013 orthologous protein domains that are informative for virus taxonomy. Using this approach, viral contigs can be rapidly classified against the taxonomy associated with the ViPhOGs, regardless of the type of virus (eukaryotic, bacterial, archaeal) or genome type (DNA/RNA), as long as there are representative ViPhOGs in the database.

This leads me into my main comment on the manuscript and pipeline, the taxonomy itself is outdated and wrong. On lines 209-2216, the authors write that the Virus-Host DB was used in November 2019 and the NCBI Taxonomy in March 2020. By the time this paper is published, the taxonomy will be at least four years out of date. The NCBI Taxonomy in March 2020 still used the taxonomy release issued by the International Committee on Taxonomy of Viruses in 2019, Master Species List MSL#34 (https://ictv.global/taxonomy/history) which contains 14 orders, 150 families, 79 subfamilies, 1019 genera and 5560 species. In contrast, the current taxonomy release (MSL#37) comprises additional higher order taxa: 6 realms, 10 kingdoms, 17 phyla, 39 classes, 65 orders, 233 families, 2606 genera and 10434 species (Walker et al, 2022, table 1, https://link.springer.com/article/10.1007/s00705-022-05516-5/tables/1). Importantly, this taxonomy release saw the removal of the bacteriophage families Myoviridae, Podoviridae and Siphoviridae which were not monophyletic, and the removal of the order Caudovirales in favour of the class Caudoviricetes, the taxa most commonly recovered from metagenome studies (see Fig 3 and 5 for example). In addition, many new families of archaeal viruses have been established since 2019, and entirely new phyla have been defined with viruses that had never before been classified.

Since virus taxonomy is dynamic when new sets of viruses are discovered, any pipeline that claims to provide a taxonomic classification of viruses needs to be able to incorporate yearly updates to its taxonomy, or make clear disclaimers that the taxonomic classification is not up-to-date. In the case of VIRify, I presume updating the taxonomy could be achieved in a relatively straightforward way by remapping the ViPhOG database to the latest taxonomy database.

A second concern related to the taxonomic assignments, is that this pipeline does not seem to have been tested or validated using existing complete virus genomes, only using mock community assemblies which could have many incomplete genome fragments (lines 360-363). In my opinion, it is important to understand how accurate the taxonomic predictions are in the most ideal situation, in order to understand the limitations of the pipeline.

As part of this review, I have installed and run the pipeline on some test data and will be providing observations and some recommendations.

Installation: I used an ubuntu virtual machine and used the nextflow install as was suggested on the GitHub page. Installation was seamless and quick. I tried a similar installation locally on a Mac using the terminal and ran into versioning issues with nextflow and the virify pipeline and abandoned this route.

Running pipeline: I used a mock dataset of 5 complete bacteriophage genomes which were all similar to well-characterised phages but not necessarily identical. The pipeline ran very quickly and easily. I ran a second set of ~5000 contigs from a virome which finished overnight. No issues.

Results of pipeline:

Small dataset:

Four out of the five bacteriophages, including the reference RNA phage MS2 were categorised as low confidence predictions in the first step of the pipeline. For me, this calls into question the thresholds that were used to categorise high confidence versus low confidence. Could the authors benchmark the virus detection categorisation with a set of known viruses from GenBank to see if the distinction between high-confidence and low-confidence needs to be made and what the best thresholds should be?

Taken into account the outdated taxonomy database used, the taxonomic assignments of the phages were correct at the genus level for only 2 out of 5 phages. MS2 was correctly classified at the family level but no genus level classification. One phage was correctly assigned at the order level and one was unclassified. No incorrect assignments were made. However, low sensitivity at the genus and family level means that there is some optimisation required for the “voting system” or potentially an update to the ViPhOG database itself to include a more up-to-date set. For the smaller viruses, a threshold for at least two hits at the genus level (lines 352-353) may be too stringent. For example, most members of the phylum Cressdnaviricota only encode two proteins.

Larger dataset:

Of the ~5000 contigs that I considered potentially viral based on an internal dataset using VirSorter2 and VIBRANT, 921 were considered viral using VIRify. The decision to minimise false positives therefore may have a consequences for an increased amount of false negatives. I consider this a feature, rather than a bug, because it’s a decision that needs to be made in viromics research. However, I do suggest that the authors be clear about this in the manuscript and pipeline.

The taxonomic assignments that were made were largely unclassified viruses (780) and the rest of them were assigned to the order Caudovirales. The family, subfamily and genus assignments appeared to be correct.

Overall, I found the pipeline easy to work with and its outputs easy to understand and use. My assessment is that it favours specificity over sensitivity in both the virus detection and taxonomic assignments.

In terms of the manuscript, my final comment is that the literature/introduction and discussion should be updated to account for pipelines and new tools that have been developed. The authors have already indicated that they may update the tool VirSorter to VirSorter2, but there is also an update of VirFinder to DeepVirFinder and new tools such as VIBRANT that have been developed since the authors started working on VIRify. Similarly, other pipelines have been published, for example MetaPhage and Hecatomb, and potentially others. I do not expect these to be benchmarked but they should at least be acknowledged in the introduction or discussion.

**Have the authors made all data and (if applicable) computational code underlying the findings in their manuscript fully available?**

Reviewer #1: Yes

Reviewer #2: Yes

Reviewer #3: Yes

PLOS authors have the option to publish the peer review history of their article (what does this mean?). If published, this will include your full peer review and any attached files.

Reviewer #1: No

Reviewer #2: No

Reviewer #3: No
---

## [Decision Letter · Decision Letter 1]

9 Aug 2023

Dear Dr Rangel-Piñeros,

We are pleased to inform you that your manuscript 'VIRify: an integrated detection, annotation and taxonomic classification pipeline using virus-specific protein profile hidden Markov models' has been provisionally accepted for publication in PLOS Computational Biology.

Best regards,

Christos A. Ouzounis

Academic Editor

PLOS Computational Biology

Thomas Leitner 

Section Editor

PLOS Computational Biology

Reviewer's Responses to Questions

**Comments to the Authors:**

Reviewer #2: Authors have adequately addressed all of the questions that I've raised.

I do recommend, however, reading-through newly added sections to improve readability/grammar. Here are a couple of examples of sentences that "sound off" :

lines 886-888 - VIRify is a ... resource that joins a group of tools ... and **that** can be broadly classified as protein similarity-based methods -> maybe change to ... and are broadly classified as ...

Line 897, the advantage of using protein profile HMMs is that they allow **the gathering** of viral reference data in a more efficient and compact way - "gathering of viral references" - does it simply mean that viral reference data for protein profile HMMs can be stored in a more compact way ?

Reviewer #3: No further comments.

**Have the authors made all data and (if applicable) computational code underlying the findings in their manuscript fully available?**

Reviewer #2: Yes

Reviewer #3: Yes

PLOS authors have the option to publish the peer review history of their article (what does this mean?). If published, this will include your full peer review and any attached files.

Reviewer #2: No

Reviewer #3: No

---

## [Editor Report · Acceptance letter]

24 Aug 2023

PCOMPBIOL-D-22-01555R1 

VIRify: an integrated detection, annotation and taxonomic classification pipeline using virus-specific protein profile hidden Markov models

Dear Dr Rangel-Piñeros,

I am pleased to inform you that your manuscript has been formally accepted for publication in PLOS Computational Biology. Your manuscript is now with our production department and you will be notified of the publication date in due course.

With kind regards,

Zsofi Zombor
